# Effects of High-Lipid Dietary Protein Ratio on Growth, Antioxidant Parameters, Histological Structure, and Expression of Antioxidant- and Immune-Related Genes of Hybrid Grouper

**DOI:** 10.3390/ani13233710

**Published:** 2023-11-30

**Authors:** Weibin Huang, Hao Liu, Shipei Yang, Menglong Zhou, Shuang Zhang, Beiping Tan, Yuanzhi Yang, Haitao Zhang, Ruitao Xie, Xiaohui Dong

**Affiliations:** 1Laboratory of Aquatic Nutrition and Feed, College of Fisheries, Guangdong Ocean University, Zhanjiang 524088, China; hwbstudy@163.com (W.H.);; 2Guangdong Engineering Technology Research Center of Aquatic Animals Precision Nutrition and High Efficiency Feed, Zhanjiang 524088, China; 3Key Laboratory of Aquatic, Livestock and Poultry Feed Science and Technology in South China, Ministry of Agriculture and Rural Affairs, Zhanjiang 524000, China

**Keywords:** hybrid grouper, aquaculture, animal feed, liver health, oxidative stress

## Abstract

**Simple Summary:**

In this study, the effects of different protein levels on the growth performance, antioxidant levels and immunity of hybrid groupers on a high-fat diet were investigated. Conclusion: An appropriate protein level can improve the growth performance of groupers, increase the activity level of antioxidant enzymes in serum and liver, and increase the expression level of antioxidant and immune genes of hybrid groupers. This provides valuable data to support the application of high-lipid diets in the aquaculture industry, allowing the feed industry to reduce the protein level for groupers and save costs.

**Abstract:**

The hybrid grouper (♀ *Epinephelus fuscoguttatus* × ♂ *E*. *lanceolatus*) is a new species of grouper crossed from giant grouper (*E. lanceolatus*) as the male parent and brown-marbled grouper (*E. fuscoguttatus*) as the female parent. We hypothesized that optimal levels of dietary protein may benefit liver function. High-lipid diets are energetic feeds that conserve protein and reduce costs, and are a hot topic in aquaculture today. Therefore, the objective of the research is to investigated the effects of dietary protein level in high-lipid diets on serum and liver biochemistry, liver histology, and liver immune and antioxidant indexes and gene mRNA expression of the juvenile hybrid grouper (♀ *Epinephelus fuscoguttatus* × ♂ *E. lanceolatus*). Six iso-lipidic (161 g/kg) diets were formulated containing graded levels of protein (510 as control, 480,450, 420, 390 and 360 g/kg). Each treatment consisted of three replicates and 30 fish (6.70 ± 0.02 g) in one replicate. After an 8-week feeding experiment, the results indicated the following: (1) With the decreasing of dietary protein level, the specific growth rate (SGR) of groupers increased gradually and then decreased; SGRs of the 390 and 360 g/kg groups were significantly lower than other groups (*p* < 0.05). (2) In terms of serum and liver, the activity of antioxidant enzymes such as catalase (CAT) and superoxide dismutase (SOD), and the total antioxidant capacity (T-AOC) content, and the activity of immune enzymes such as lysozyme (LYS) and immunoglobulin (IgM) was significantly increased under the appropriate protein level. (3) Based on liver histology, we know that high or low dietary protein levels cause liver damage. (4) Dietary protein levels can significantly affect the mRNA expression levels of an anti-inflammatory factor gene (*tgfβ*), pro-inflammatory factor genes (*il6*, *il8*), heat shock proteins, and antioxidant and immune genes (*hsp70* and *hsp90*, *gpx*, *nrf2*, *keap1*). It is concluded that the appropriate protein level can promote the growth performance of groupers, improve antioxidant activity and immune enzyme activity in serum and liver, and enhance the expression of immune genes.

## 1. Introduction

The hybrid grouper (♀ *Epinephelusfuscoguttatus* × ♂ *E. lanceolatus*) is a new species of grouper crossed from giant grouper (*E. lanceolatus*) as the male parent and brown-marbled grouper (*E. fuscoguttatus*) as the female parent, which belongs to the *Perciformes*, Suborder *Perciformes*, *Serranodae* and Grouper [1,2]. It is favored by consumers and farmers due to its fast growth rate, strong disease resistance and high return on agricultural investment, and has a large consumer market in southern China [3]. These huge development backgrounds have made it a hot spot for scientific research in aquaculture, reaching up to 204,100 tons in 2021 [3].

Protein is the most expensive major component of formulated feed and plays a key role in fish growth performance as the rate of protein deposition determines growth rate [4,5]. It has been shown that both low and high levels of dietary protein inhibit fish growth and disrupt tissues and organs, while optimal levels of dietary protein are critical for maximum growth and improved health [6,7]. On the other hand, when dietary protein levels exceed requirements, the excess is catabolized for energy production, which is not only an economic waste but also increases the amount of ammonia released, resulting in the degradation of rearing water [8]. Therefore, knowing the dietary protein requirements of farmed fish species is beneficial from both economic and environmental perspectives.

Lipids are an essential nutrient for animal growth and play a vital role in the metabolism and immunity of organisms [9]. Lipids play an important role in aquafeeds by providing essential fatty acids and energy for the growth and maintenance of health in aquatic animals [10]. At the same time, higher lipids can serve to save protein and reduce production costs [11]. Therefore, a high-lipid diet has become a trend of application in the aquafeed industry.

In our previous study [12], we designed an experiment with a protein level of 500 g/kg and three lipid levels (60 g/kg, 120 g/kg and 160 g/kg) and concluded that the 160 g/kg group promoted grouper growth but affected liver health. It was taken into account that lipids are used as an energy source, that they save a certain amount of protein and that excess protein increases the liver load of the fish [13]. In addition, previous studies have shown that the protein requirement of groupers is 50% or more [13], and the effect of high lipids on the protein requirement of groupers is not yet known. Therefore, it is of great importance to study the effect of dietary protein levels in a high-lipid background on hybrid groupers. This is a data theory of precise nutrition for the hybrid grouper, and has guiding significance for the application of high-lipid diets in the feed industry.

## 2. Materials and Method

### 2.1. Experimental Conditions

The experimental fish were purchased from Hong-Yun farm in Dongjian town (Zhanjiang, China) and transported to the Marine Biological Research Base of Guangdong Ocean University (Zhanjiang, China). About 810 fish were temporarily reared in a cement pond of dimensions 5 m (length) × 4 m (width) × 1.8 m (depth) and fed a commercial diet (50% protein level, 10% lipid level, Haida Aquatic Diet Co., Ltd., Zhanjiang, China) for 2 weeks until they reached the size required for the experiment (6.70 ± 0.02 g). In total, 540 fish were randomly allocated to 18 fiberglass tanks (500 L), each provided with one piece of polyvinylchloride (PVC) pipe of 20.0 cm (diameter) × 30.0 cm (length) as a shelter for the fish. Each feed was given to fish in three tanks twice a day (8:00 and 16:00) until apparent satiation was observed. The daily food intake was recorded for 8 weeks. About 70% of the water was changed every day to maintain water quality. During the breeding experiment, the water conditions were temperature 29–32 °C, salinity 28–30, dissolved oxygen > 7 mg/L and nitrates < 0.05 mg/L, which were tested by a multi-parameter water quality detector (PTF-001B, WBD Biotechnology Co., Ltd., Zhanjiang, China), according to the method described in our previous study [14].

### 2.2. Experimental Diets

The composition and nutritional composition of the experimental diets are shown in Table 1. Six isolipid diets with different protein levels (510 as control, 480, 450, 420, 390 and 360 g/kg) were formulated. All ingredients were crushed and passed through a 60 mesh sieve, then thoroughly mixed using the progressive enlargement method, as described by Long et al. [15]. After homogenization of the feed, mix with distilled water until a wet dough is obtained, which does not fall apart when held firmly in the hands. The diets were processed into 2. 0 mm and 2.5 mm diameter pellets by a twin screw extruder (F-26, South China University of Technology, Zhanjiang, Guangdong Province, China), air-dried at room temperature then ground and sieved to an appropriate size and stored in ziploc bags at −20 °C until use [1].

### 2.3. Sample Collection

At the end of the trial, the fish were fasted for 24 h. Fish were anesthetized with eugenol (1:10,000), and all fish in each tank were weighed and counted to calculate specific growth rate (SGR) and feed conversion ratio (FCR). Then, six fish from each tank were randomly selected for blood collection using 1 mL sterile syringes. Blood was placed in 1.5 mL microcentrifuge tubes and stored at 4 °C for 12 h. The blood was later centrifuged (4000 rpm for 15 min at 4 °C) and the serum collected and stored at −20 °C for enzyme activity analysis. The liver was separated for enzyme activity analysis. The livers of 2 fish in each tank were kept in 4% formalin solution for histological analysis. Then, two fish were randomly selected from each tank for rapid isolation of liver and placed in the enzyme-free EP tube with RNA for later gene expression analysis and stored in liquid nitrogen.

### 2.4. The Methods of Analysis

The calculation formulas of growth parameters and morphological indices were as follows [16]: specific growth rate (SGR, %) = 100 × (ln final weight − ln initial weight)/days of experiment; survival rate (SR, %) = 100 × (total number of fish at termination/total number of fish stocked); feed conversion ratio (FCR) = feed intake/(final body weight − initial body weight). The diets and whole fish were analyzed according to the official method of AOAC (1995) [17]. The moisture content was measured by drying at 105 °C to a constant weight (GB/T 6435-2006 [18]). The crude protein content was determined using the Kjeldahl method (GB/T 6432-2018 [19]). The crude lipid content was determined using the Soxhlet extraction method (GB/T 5512-2008 [20]).

### 2.5. Antioxidant and Immunity Indexes in Serum and Liver

Related indexes in serum and liver of hybrid grouper were detected by commercial ELISA kits (Shanghai Enzyme-linked Biotechnology Co., Ltd., Shanghai, China), including activities of catalase (CAT) and superoxide dismutase (SOD), and the content of malondialdehyde (MDA) and reactive oxygen species (ROS) in serum. Activities of SOD, CAT, alkaline phosphatase (AKP), acid phosphatase (ACP), lysozyme (LYS), aspartate transaminase (AST) and alanine transaminase (ALT), and the content of MDA, ROS, complement C3, complement C4 and immunoglobulin M (IgM) in liver were recorded. Total antioxidant capacity (T-AOC) in serum and liver was checked by the kit (DPPH method, Shanghai Enzyme-linked Biotechnology Co., Ltd., Shanghai, China). All index measurements were carried out in strict accordance with the kit instructions, following a previously described method of Yin et al. [21].

### 2.6. Hepatic Histopathology

Histopathological analysis of hepatic samples was conducted by hematoxylin–eosin staining, as [22] described. After fixation with 4% paraformaldehyde, the liver tissue was dehydrated with ethanol gradient and embedded in paraffin. The embedded liver tissue was sectioned into 5 μm thick slices and observed under a microscope (Hitachi H-7650 transmission electron microscope, Hitachi Ltd., Tokyo, Japan) [22].

### 2.7. RNA Extraction and Real-Time Quantitative Polymerase Chain Reaction (RT-qPCR)

Transzol UP (TransGen Biotech, Beijing, China) of 1 mL was added to the samples and the total RNA was extracted according to the manufacturer’s protocol. The quantity and quality of isolated RNA were detected at 260 nm and 280 nm using a NanoDrop 2000 spectrophotometer (Gene Company Limited, Guangzhou, China) and electrophoresis in 1% agarose gel, respectively. The first-strand cDNA was synthesized using PrimeScript^TM^ RTreagent Kits with cDNA Eraser (Takara, Japan) according to the manufacturer’s instructions. The cDNA was stored at −20 °C for real-time quantitative polymerase chain reaction (RT-qPCR). RT-qPCR was performed in a 384-well plate with a 10 µL reaction volume containing 5 µL of SYBR^®^ Green Real-time PCR Master Mix, 0.8 µL of each primer, 1 µL of cDNA sample and 3.2 µL of RNase Free ddH_2_O. The PCR conditions were set using a thermal programmer at 95 °C for 30 s, 40 cycles of 95 °C for 5 s and 60 °C for 34 s. Each sample was tested in triplicate. Primers of the reference gene (β-actin) and target gene were designed according to published sequences of groupers (Table 2). Threshold cycle (Ct) values were collected from each sample after finishing the process. The relative expression levels were calculated using the 2^−ΔΔCt^ method [12].

### 2.8. Statistical Analysis

The results were expressed as mean ± standard deviation (mean ± SEM) and one-way analysis of variance (one-way ANOVA) was used to test the significance using SPSS version 20.0 (SPSS Inc., Michigan Avenue, Chicago, IL, USA); Tukey’s multiple comparison method was used to compare the data. The differences between all test results were considered significant at *p* < 0.05. Diet group was the variable and the rest of the experimental indicators such as growth performance, enzyme activity and genes were the dependent variables.

## 3. Results

### 3.1. Growth Performance

In this study, with the decreasing of dietary protein level, the SGR of groupers increased gradually and then decreased significantly (Table 3); the SGRs of the 390 and 360 g/kg groups were significantly lower than those of other groups (*p* < 0.05) except the 420 g/kg group. The dietary protein level had a significant effect on the FCR of groupers; the levels of the 360 g/kg and 390 g/kg groups were higher than other groups (*p* < 0.05). The SR of the 360 g/kg group was significantly lower than other groups (*p* < 0.05) and there was no significant difference in SR among the other groups. Based on SGR, the second-order polynomial regression analysis showed that the optimum dietary protein level was 430.36 g/kg (Y = 0.0017x − 0.8467, R^2^ = 0.9919; Y = −0.0005x + 3.9733, R^2^ = 0.9643) (Figure 1).

### 3.2. Antioxidant Parameters in Serum

As shown in Table 4, the catalase (CAT) activity in the serum of the groupers fed the 420 g/kg diet significantly increased compared with the 510 g/kg group (*p* < 0.05), and no statistically significant difference was observed between the other trial groups and the control group. The serum superoxide dismutase (SOD) levels of the 450 g/kg, 420 g/kg, 390 g/kg and 360 g/kg groups were significantly higher than the control group (*p* < 0.05). The total antioxidant capacity (T-AOC) of serum in the 420 g/kg and 390 g/kg groups were significantly higher than those of the control group (*p* < 0.05). Compared with the 510 g/kg group, the reactive oxygen species (ROS) levels in the 420 g/kg and 390 g/kg groups were significantly reduced (*p* < 0.05), but the malondialdehyde (MDA) content in the serum of groupers was not affected by the protein level in the high-lipid diet (*p* > 0.05).

### 3.3. Antioxidant and Non-Specific Immunity Parameters in Liver

As shown in Table 5, hepatic CAT activity in the 390 g/kg group reached the maximum and was significantly higher than that of the control group (*p* < 0.05), but had no significant difference from the other groups (*p* > 0.05). The SOD, LYS and AKP activities, and T-AOC and IgM content of the 420 g/kg group were significantly higher than those of the control group (*p* < 0.05). Compared to the control group, dietary protein level had no significant effects on ROS, MDA and acid phosphatase (ACP) (*p* > 0.05), and a significant reduction in aspartate transaminase (AST) and alanine transaminase (ALT) was found in the 390 g/kg group (*p* < 0.05).

### 3.4. Histological Structure of Liver

Hepatic histological observation is summarized in Figure 2 (stained by hematoxylin and eosin). Liver tissue of the 510 g/kg group showed an indistinct cellular outline and large hepatocellular vacuolation. And these pathological symptoms still appeared in the 480 g/kg and 450 g/kg groups. With the decrease in protein level (420 g/kg groups), cellular outline gradually returned to normal and hepatocellular vacuolation distinctly reduced. With a further reduction in protein levels (390 g/kg and 360 g/kg groups), hepatocytes became vacuolated and the cell outline was indistinguishable.

### 3.5. Expression of Antioxidant- and Immune-Related Genes in Liver

Figure 3 shows that dietary protein has different effects on antioxidant- and immune-related gene expression in the liver of the grouper. Dietary protein significantly up-regulated the mRNA expression of *keap1* and *gpx* in the 450 g/kg and 420 g/kg groups, and the expression of *hsp70* was significantly up-regulated in the 420 g/kg groups (*p* < 0.05). *nrf2* expression in the 420 g/kg group and *hsp90* expression in the 480 g/kg and 420 g/kg groups were significantly higher than other groups (*p* < 0.05). Compared with the control group, in the 360–450 g/kg groups, the expression of *il 6* was significantly reduced (*p* < 0.05). *il8* expression in the 450 g/kg and 420 g/kg groups was significantly down-regulated by dietary protein level (*p* < 0.05). In the protein-treated groups, the expression of *tgfβ* was significantly down-regulated (*p* < 0.05).

## 4. Discussion

Protein is essential for fish growth, providing essential amino acids and energy. It is the primary parameter for determining fish growth and formula feed performance [23,24]. In the present study, the SGR of hybrid groupers increased gradually and then decreased with the decrease in dietary protein level, and that is consistent with the results of the large-size grouper (*Epinephelus coioides*) [23], juvenile striped catfish (*Pangasianodon hypophthalmus*) [25] and Pacific white shrimp (*Litopenaeus vannamei*) [26]. Excess protein in feed is not used for fish growth but decomposes, increasing nitrogen metabolism burden and reducing fish growth [27,28,29]. Simultaneously, insufficient protein content in feed can lead to a reduction in tissue protein anabolic capacity, thus hindering animal growth [30]. Our study demonstrated a consistent and significant decrease in FCR with increasing dietary protein level, which is in agreement with studies on the European sea bass (*Dicentrarchus labrax*) [31], spotted sea bass (*Lateolabrax maculatus*) [4] and sunshine sea bass (*Morone chrysops* ♀ × *Morone saxatilis* ♂) [32]; however, the opposite has been found in the large grouper *Epinephelus coioides* [23], which may be due to the size of the fish. As just mentioned, protein deficiency affects the ability to synthesize tissue proteins and thus restrains the growth of fish [30,33], which explains the significantly lower survival rate (SR) in the low-protein group (360 g/kg group), and the same conclusion was found for the oriental river prawn (*Macrobrachium nipponense*) [34]. Based on SGR, broken-line regression analysis showed that the optimum dietary protein level in high-lipid diets in the hybrid grouper (♀ *Epinephelusfuscoguttatus* × ♂ *E. lanceolatus*) was 430.36 g/kg, while the optimum protein level for juvenile groupers was 480 g/kg [35]. The difference is attributed to the varying lipid levels in the two experiments. Lipids can be used as an energy source, enhancing protein efficiency and reducing protein expenditure, thus making high-lipid diets cost-effective [16].

Changes in serum biochemical indicators can indicate fish metabolism and physiological status, which is often used to assess fish health and environmental adaptability [36]. Fish’s antioxidant defense system is divided into two main categories: enzymatic and non-enzymatic. SOD and CAT are two important antioxidant enzymes that can effectively eliminate superoxide anion radicals (O^2−^), free oxygen (O), hydroxyl radicals (–OH), H_2_O_2_ and other reactive oxygen species from the body [37,38].

Our study revealed that the activity of SOD and CAT in serum increased and then decreased as protein levels decreased, reaching a peak in the 390 g/kg and 420 g/kg groups, respectively. Both enzymes exhibited a similar pattern of change in activity. The rise in SOD activity catalyzed the decomposition of superoxide anions, leading to an increase in H_2_O_2_ concentration in the body. The subsequent increase in CAT activity eliminated H_2_O_2_ from tissues and organs, thereby reducing damage to the body and augmenting the body’s antioxidant capacity, which is in line with the findings of prior studies [39]. T-AOC is the total antioxidant capacity of the enzymatic and non-enzymatic antioxidant systems of the fish organism, and is a comprehensive indicator of the antioxidant capacity of the organism. The level of T-AOC can directly reflect the compensatory capacity of the organism to stress, as well as the metabolic status and free radical scavenging capacity of the organism [40]. In this experiment, T-AOC followed the same trend as CAT and SOD. This indicates that T-AOC is largely influenced by the activity of these two enzymes. Studies have shown that, when large amounts of oxygen radicals are present in the body [2,41], if the antioxidant capacity of the fish’s antioxidant system is weak, it fails to scavenge ROS in time, which in turn causes oxidative damage to cells and promotes lipid peroxidation [42]. In the tissues and organs of the body, the concentration of MDA, the end product of lipid peroxidation, is not only a direct reflection of the antioxidant capacity of the tissues and organs, but also a measure of the antioxidant capacity of the body [42]. In this experiment, the concentration of MDA did not differ between the groups. ROS concentrations were the lowest in the 420 g/kg and 390 g/kg groups, and were significantly lower than in the control group. This trend was not only in serum but also in subsequent liver tissues. This result was due to the increase in SOD and CAT activities, which accelerated the clearance of ROS in fish bodies [39].

Consumption of a high-lipid diet predisposes the liver to oxidative damage and different protein levels have further effects on liver [43]. Thus, the antioxidant status of the liver determines its susceptibility to oxidative damage. The body’s ability to scavenge peroxides and free radicals, prevent their excessive accumulation and maintain an antioxidant state is mainly mediated by antioxidant enzymes [44]. Similar to the serum, SOD, CAT, T-AOC and ROS in the liver show the same trend, and this is consistent with studies in the juvenile Nile tilapia (*Oreochromis niloticus*) [45], juvenile common carps (*Cyprinus carpio*) [46] and golden pompano (*Trachinotus ovatus*, Linn) [39]. LYS and immunoglobulins are closely related to immunity in fish, where LYS is mainly derived from macrophages and is an important non-specific immune factor with destructive effects on exogenous substances, while IgM is an important immunoglobulin and antibody for fish [47,48], which functions as a specific immune factor to recognize and neutralize foreign antigens such as bacteria and viruses, and plays an important role in protection against foreign pathogens [49].

In this experiment, liver IgM and LYS achieved a maximum value in the 420 g/kg group and was significantly higher than in the control group, indicating that protein levels can affect liver immunity levels, and that too high or too low protein levels can also cause damage to the liver, which is consistent with the study of tilapia (*Oreochromis niloticus*) [50]. ACP and AKP are important regulatory enzymes in the metabolic process of the body and are often used as important regulatory enzymes for the evaluation of non-specific immune performance [51], and a decrease in their activity leads to a decrease in non-specific immune performance [52]. The present study showed that, with decreasing protein levels, hepatic AKP increased gradually and reached the maximum value in the 420 g/kg group, and then decreased, while there was no significant difference in ACP. This suggests that appropriate protein levels in feed may help to improve the disease resistance of groupers, but too high or too low protein levels may inhibit AKP activity and have a negative impact on the immune system, in line with the study on *Schizothorax o’connori* [53]. Among the hepatic transaminases and combined deamidation, transaminases play a key role as the key enzymes that catalyze the transfer of amino acids to keto acids. AST and ALT are the most important transaminases, and can indicate hepatic metabolic function and reflect the degree of liver health [54]. In this experiment, AKP activity was significantly higher in the 420 g/kg group, which was consistent with the study of the juvenile common carp (*Cyprinus carpio*) but differed from the studies of yellow croakers (*Pseudosciaena crocea*) [55,56].

Histological morphology of the liver is considered to be a good indicator of the nutritional status, metabolism and liver health of fish [57,58]. In this study, paraffin sections of liver were prepared to evaluate the efficacy of protein levels in a high-lipid diet on hepatic histologic changes. Previous studies have shown that the liver could be damaged by high-lipid diets [59] or environmental toxicants manifested by nuclear hypertrophy, karyolysis, indistinguishable cell contours, extensive hepatocellular vacuolation and other phenomena [2,60,61]. Similar symptoms were observed in the control group of this study but the 420 g/kg group reduced the histological abnormalities caused by the high-lipid diet. These results suggest that adequate levels of dietary protein improve the liver morphology of hybrid groupers fed a high-lipid diet.

Aquatic animals can adapt to stress through metabolic and biochemical reactions or by developing antioxidant defenses [62,63]. These include the formation of related molecules, the induction of antioxidant enzymes and certain changes in tissue structure. Among these, organisms can respond to stress by synthesizing different stress proteins. Heat shock proteins (HSPs) are the most critical stress proteins, and play an important role in the induction of both innate and adaptive immune responses [64,65]. HSP70 is an important component in fish that plays a molecular chaperone role in antioxidant and immune responses [66]. HSP90 is involved in the maintenance of various components of the cytoskeleton and steroid hormone receptors [44]. In the present experiment, the gene expression of *hsp70* and *hsp90* was significantly up-regulated in the 420 g/kg group (*p* < 0.05), which improved the immunity of the fish. This may indicate that appropriate dietary protein levels can promote the accumulation of *hsp70* and *hsp90* in fish to improve disease resistance or immunity, and this is consistent with the study presented on the blunt snout bream (*Megalobrama amblycephala*) [67].

GSH-Px can convert toxic peroxides to non-toxic hydroxyl compounds, thus protecting the structure and function of cell membranes from interference and damage caused by peroxides. As key players in the defense against oxidative damage, nuclear factor-erythroid-2-associated factor 2 (*nrf2*) and kelch-like epichlorohydrin-associating protein 1 (*keap1*) are involved in the regulation of antioxidant genes [60,68,69]. Under normal conditions, most *nrf2* molecules bind to *keap1* and are maintained at low levels by ubiquitination and proteasomal degradation of *keap1*; however, under oxidative stress, *nrf2* is released from *keap1*, enters the nucleus, activates transcription and binds to downstream antioxidant response elements to activate the cytoprotective expression of antioxidant enzyme genes [70]. In the current study, up-regulated antioxidant gene (*gpx*) expression was positively related to *nrf2* expression, indicating that an optimal protein level enhanced the antioxidant capacity to resist the oxidative stress induced by the high-lipid diet, probably through activation of the nrf2-keap1 signal pathway. At the same time, we reasonably hypothesized that this gene pathway influenced the antioxidant performance of the fish, combining the above serum and liver enzyme activity, and hepatic histological structure.

In addition to causing hepatic steatosis and oxidative damage, a high-lipid diet can also induce hepatic inflammatory responses [44]. Previous studies have shown that variations in dietary protein levels could affect the liver function and immune condition of different aquatic animals [71,72]. *il6* and *il8* are important intracellular pro-inflammatory cytokines; *tgfβ* is an anti-inflammatory cytokine that is important for improving tissue resistance to disease and inhibiting inflammatory responses [58,73]. In this study, appropriate levels of protein in a high-lipid diet significantly increased the mRNA expression of the anti-inflammatory factor *tgfβ* and significantly decreased the mRNA expression of the pro-inflammatory factors *il6* and *il8* compared to the control group, and the mRNA expression of *il6* and *il8* was significantly down-regulated in the 390 g/kg and 420 g/kg groups, and the opposite was true for *tgfβ* gene expression. Previous studies have shown that *il6*, *il8* and *tgfβ* are important factors in the classical NF-κB pathway [74,75]. Taken together, this suggests that optimal levels of dietary protein may reduce liver inflammation in the grouper and regulate the expression of inflammatory factors through the NFκB pathway, consistent with previous studies on the yellow catfish (*Pelteobagrus fulvidraco*) [54] and bighead carp (*Aristichthys nobilis*) [71].

## 5. Conclusions

In this study, we showed that appropriate protein levels can promote the growth performance as well as increase the antioxidant enzyme activity of hybrid groupers in serum and liver, and can modulate the effect of enzyme activity by up-regulating antioxidant genes and immune-related genes. Suitable protein levels can influence the healthiness of liver tissues. We recommend an appropriate protein level of 43–45% at the 16% lipid level.

## Figures and Tables

**Figure 1 animals-13-03710-f001:**
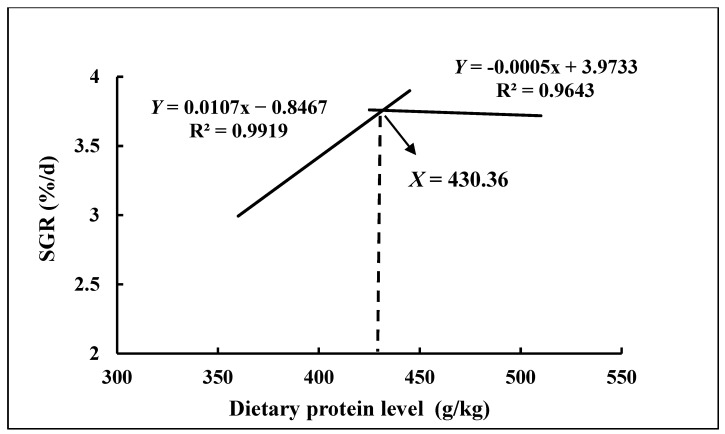
Relationship between dietary protein level and specific growth rate of groupers.

**Figure 2 animals-13-03710-f002:**
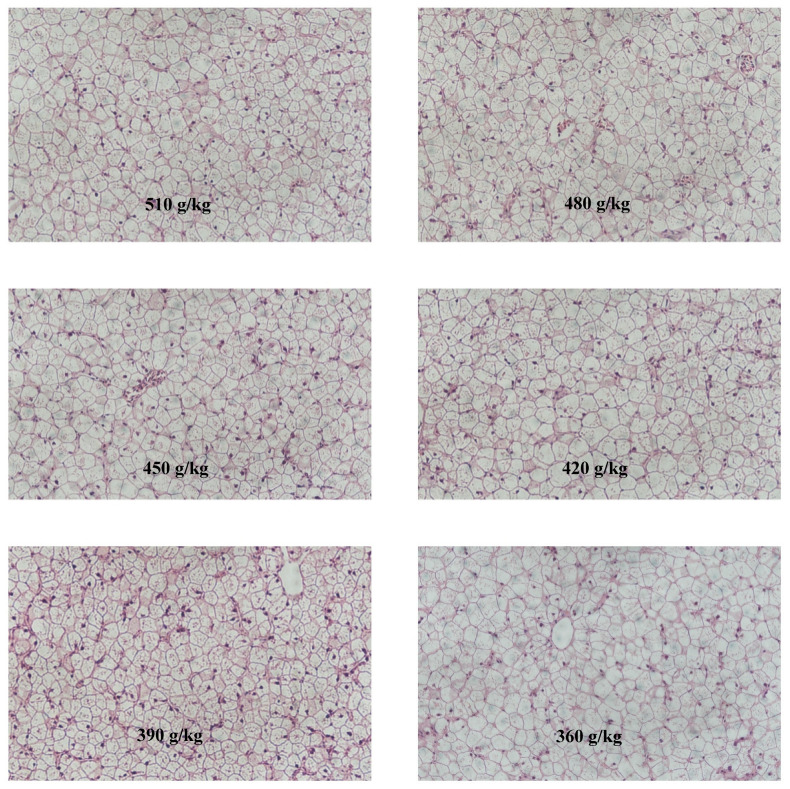
Hepatic histological structures of hybrid groupers fed high-lipid diets with different protein levels (hematoxylin and eosin (H & E) × 200).

**Figure 3 animals-13-03710-f003:**
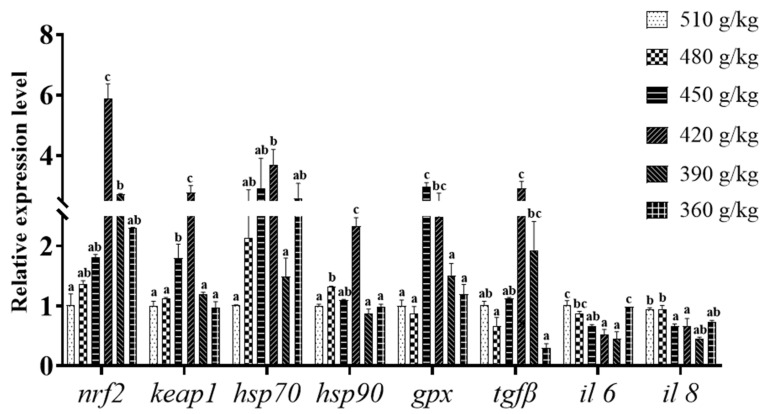
Effect of dietary protein levels on relative expression of antioxidant and immune genes in liver of hybrid grouper. Values are means ± SEM (*n* = 3). Different letters assigned to the bars represent significant differences using Tukey’s test (*p* < 0.05). *hsp70*, heat shock protein 70; *hsp90*, heat shock protein 90; *gpx*, glutathione peroxidase; *nrf2*, nuclear factor erythroid related factor 2; *tgfβ*, transforming growth factor β; *keap1*, kelch-like epichlorohydrin-associating protein 1; *il6*, interleukin 6; *il8*, interleukin 8.

**Table 1 animals-13-03710-t001:** The formula and proximate composition of the basal diet (dry matter basis, %).

Ingredients	Dietary Protein Level (%)					
	51	48	45	42	39	36
Fish meal	43	39	35	31	27	23
Wheat gluten powder	5	5	5	5	5	5
Wheat flour	17	17	17	17	17	17
CAP	15	15	15	15	15	15
Soybean lecithin	1.5	1.5	1.5	1.5	1.5	1.5
Fish oil	5	5	5	5	5	5
Corn oil	7	7.28	7.57	7.86	8.15	8.44
Dextrinized starch	2	2	2	2	2	2
Premix ^a^	1	1	1	1	1	1
Vitamin C	0.05	0.05	0.05	0.05	0.05	0.05
Choline chloride	0.5	0.5	0.5	0.5	0.5	0.5
Ca(H_2_PO_4_)^2^	1.5	1.5	1.5	1.5	1.5	1.5
Ethoxyquin	0.1	0.1	0.1	0.1	0.1	0.1
Attractant	0.15	0.15	0.15	0.15	0.15	0.15
Sodium carboxymethyl cellulose	1	1	1	1	1	1
Cellulose microcrystalline	0.2	3.92	7.63	11.34	15.05	18.76
Lysine	0.00	0.16	0.31	0.47	0.62	0.78
Methionine	0.00	0.06	0.12	0.18	0.24	0.30
Proximate composition ^b^						
Crude protein	50.9	47.61	44.47	41.15	38.79	35.77
Crude lipid	16.04	16.13	16.15	16.25	15.8	16.4

^a^ Premix provided by Qingdao Master Biotechnology Co., Ltd. (Qingdao, China). ^b^ Measured value.

**Table 2 animals-13-03710-t002:** Primers used in RT-qPCR.

Primer Names	Forward and Reverse Primer Sequence (5′ to 3′)	GenBank Accession No.
*β-action-F* /R	GGCTACTCCTTCACCACCACA /TCTGGGCAACGGAACCTCT	AY510710.2
*hsp70-*F /R	AAAAGCACGGCTCATACTCAC /GCTCTGGCAGCAGTGGAA	FJ196235.1
*hsp90-*F /R	GCTACAGAGCAGCACGACA /CTCCTCATCTTCGCTTTCC	DQ232867.1
*keap1-*F /R	GAAGTCTGAGAGGGAGAGCGGTAG /GTGTGACAGGGTTGAGCGATGAG	XM-033623805.1
*gpx-*F /R	TCCTCTGTGGAAGTGGCTGA /TCATCCAGGGGTCCGTATCT	HQ441085.1
*il-6-*F /R	AGGAAGTCTGGCTGTCAGGA /GCCCTGAGGCCTTCAAGATT	JN806222.1
*il-8-*F /R	AAGTTTGCCTTGACCCCGAA /AAGCAGATCTCTCCCGGTCT	GU988706.1
*tgfβ-*F /R *nrf2-*F /R	CGATGTCACTGACGCCCTGC /AGCCGCGGTCATCACTTATC GAAGGAGCGTCTGTTGAGTGA /GAAGATGCTGCCGTTAGTTGA	GQ205390.1 KU892416.1

Notes: *hsp70, heat shock protein 70; hsp90, heat shock protein 90; gpx, glutathione peroxidase; nrf2, nuclear factor erythroid related factor 2; tgfβ, transforming growth factor β; keap1, kelch-like epichlorohydrin-associating protein 1; il6, interleukin 6; il8, interleukin 8.*

**Table 3 animals-13-03710-t003:** Growth performance of groupers fed high-lipid diets with different protein levels.

	Dietary Protein Levels (g/kg)						
Diets	510	480	450	420	390	360	*p* Value
SGR (%/d)	3.72 ± 0.03 ^a^	3.73 ± 0.06 ^a^	3.75 ± 0.05 ^a^	3.65 ± 0.02 ^a^	3.28 ± 0.02 ^b^	3.01 ± 0.07 ^c^	0.03
FCR	0.76 ± 0.01 ^a^	0.80 ± 0.05 ^a^	0.81 ± 0.01 ^a^	0.84 ± 0.01 ^ab^	1.15 ± 0.02 ^b^	1.14 ± 0.06 ^b^	0.01
SR (%)	94.44 ± 2.94 ^a^	92.22 ± 2.94 ^a^	96.67 ± 1.93 ^a^	96.67 ± 1.93 ^a^	91.67 ± 2.89 ^a^	81.11 ± 2.94 ^b^	0.01

Notes: Values in the table are means ± SEM (*n* = 3). Values in the same row with the same superscript letter or absence of superscripts are not significantly different (*p* > 0.05). SGR, specific growth rate; FCR, feed conversion ratio; SR, survival rate.

**Table 4 animals-13-03710-t004:** Serum antioxidant parameters of hybrid grouper.

Diets	SOD (mg/mL)	CAT (mg/mL)	T-AOC (U/mL)	ROS (U/mL)	MDA (mg/mL)
510	6.22 ± 0.29 ^a^	11.40 ± 0.53 ^a^	13.42 ± 0.70 ^a^	490.50 ± 11.80 ^b^	7.30 ± 0.72
480	9.03 ± 0.34 ^ab^	12.50 ± 0.66 ^ab^	14.80 ± 0.62 ^ab^	434.95 ± 13.43 ^ab^	6.76 ± 0.68
450	10.25 ± 1.04 ^b^	15.52 ± 0.85 ^ab^	16.11 ± 0.87 ^abc^	438.02 ± 15.32 ^ab^	6.63 ± 1.08
420	10.11 ± 0.84 ^b^	17.01 ± 1.46 ^b^	19.58 ± 0.19 ^c^	319.28 ± 5.83 ^a^	4.98 ± 0.52
390	14.59 ± 0.94 ^c^	14.95 ± 0.18 ^ab^	18.54 ± 0.08 ^bc^	317.93 ± 49.56 ^a^	5.54 ± 0.51
360	10.53 ± 0.34 ^b^	14.73 ± 1.02 ^ab^	16.52 ± 1.26 ^abc^	332.43 ± 45.29 ^ab^	5.60 ± 0.81
*p* value	<0.01	0.01	0.01	0.02	0.40

Notes: Values in the table are means ± SEM (*n* = 3). Values in the same column with the same superscript letter or absence of superscripts are not significantly different (*p* > 0.05). ROS, reactive oxygen species; T-AOC, total antioxidant capacity; CAT, catalase; MDA, malondialdehyde; SOD, superoxide dismutase.

**Table 5 animals-13-03710-t005:** Hepatic antioxidant and non-specific immunity parameters of hybrid grouper.

	Dietary Protein Levels (g/kg)						
Diets	510	480	450	420	390	360	*p* Value
SOD (ng/mg.pro)	13.40 ± 0.59 ^a^	14.43 ± 1.03 ^a^	12.92 ± 0.90 ^a^	20.38 ± 1.42 ^b^	16.65 ± 0.22 ^ab^	14.89 ± 0.98 ^a^	0.01
CAT (ng/mg.pro)	15.95 ± 1.58 ^a^	17.53 ± 0.75 ^a^	22.26 ± 3.78 ^ab^	23.20 ± 1.03 ^ab^	26.30 ± 0.96 ^b^	21.83 ± 1.48 ^ab^	0.01
T-AOC (U/mg.pro)	17.70 ± 1.39 ^a^	18.10 ± 1.11 ^a^	18.10 ± 0.37 ^a^	27.30 ± 3.66 ^b^	22.55 ± 1.18 ^ab^	17.80 ± 0.64 ^a^	0.01
ROS (U/mL)	333.12 ± 22.00	316.72 ± 15.06	287.33 ± 86.73	217.53 ± 33.95	192.58 ± 35.37	221.30 ± 23.76	0.08
IgM (μg/mg.pro)	25.29 ± 0.69 ^a^	36.92 ± 0.67 ^ab^	32.53 ± 4.12 ^ab^	43.29 ± 1.80 ^b^	39.06 ± 0.72 ^ab^	36.98 ± 3.34 ^ab^	0.01
LYS (mU/mg.pro)	3.28 ± 0.47 ^a^	4.35 ± 0.49 ^ab^	5.35 ± 0.61 ^ab^	6.52 ± 1.00 ^b^	6.36 ± 0.24 ^b^	4.46 ± 0.58 ^ab^	0.03
LYS (mU/mg.pro)	3.28 ± 0.47 ^a^	4.35 ± 0.49 ^ab^	5.35 ± 0.61 ^ab^	6.52 ± 1.00 ^b^	6.36 ± 0.24 ^b^	4.46 ± 0.58 ^ab^	0.03
ACP (mU/mg.pro)	12.96 ± 0.68	11.5 ± 0.75	11.46 ± 1.42	10.66 ± 1.12	13.41 ± 0.74	9.83 ± 0.57	0.11
AKP (miU/mg.pro)	11.72 ± 0.81 ^a^	12.03 ± 1.31 ^a^	12.37 ± 1.84 ^a^	19.3 ± 1.09 ^b^	16.16 ± 1.53 ^ab^	11.7 ± 1.73 ^a^	0.01
AST (mU/mg.pro)	25.52 ± 0.64 ^b^	23.48 ± 0.44 ^b^	24.77 ± 1.17 ^b^	22.28 ± 0.72 ^ab^	18.91 ± 1.23 ^a^	18.55 ± 0.53 ^a^	<0.01
ALT (mU/mg.pro)	13.22 ± 0.07 ^b^	11.65 ± 0.99 ^ab^	10.82 ± 0.92 ^ab^	10.07 ± 0.29 ^ab^	9.58 ± 0.20 ^a^	11.5 ± 0.34 ^ab^	0.04

Notes: Values in the table are means ± SEM (*n* = 3). Values in the same row with the same superscript letter or absence of superscripts are not significantly different (*p* > 0.05). ROS, reactive oxygen species; T-AOC, the total antioxidant capacity; CAT, catalase; MDA, malondialdehyde; SOD, superoxide dismutase; IgM, immunoglobulin M; LYS, lysozyme; AKP, alkaline phosphatase; ACP, acid phosphatase; AST, aspartate transaminase; ALT, alanine transaminase.

## Data Availability

The data that support the findings of this study are available on request from the corresponding author. The data are not publicly available due to privacy or ethical restrictions.

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
