# Peer review of "Effects of High-Lipid Dietary Protein Ratio on Growth, Antioxidant Parameters, Histological Structure, and Expression of Antioxidant- and Immune-Related Genes of Hybrid Grouper"

_animals, 2023, doi:10.3390/ani13233710_

Round 1

Reviewer 1 Report

Comments and Suggestions for Authors

Brief summary  

Dear Authors: 

Following the COPE ethical guidelines for peer reviewers, I am submitting the review report of the paper: 

Ameliorative effect of appropriate dietary protein levels on oxi-dative stress of hybrid groupers (♀ Epinephelus fuscoguttatus × â™‚ E. lanceolatus) induced by high-lipid diet.

Which I have mimosely read in its entirety, thanking the authors for their effort and dedication in its elaboration and consider Animals journal for review and possible publication.

The Manuscript's objective was:

Study the effect of dietary protein levels in a high lipid background on hybrid grouper.

The submitted work demonstrates in the introduction section the relevance and pertinence of the work, the species has been little explored in terms of the research objective, the research highlights the findings of Growth performance, Antioxidant parameters in serum, Histological structure of live, expression of antioxidant and immune-related genes in liver. 

The results showed that the optimal dietary protein level of juvenile hybrid grouper under the background of high lipid diet was .36 g/kg. Excessive dietary protein level could not accelerate the growth rate, and low protein level(360 g/kg) would inhibit the growth rate and survival rate.

The importance of the developed work implies that there is a knowledge gap in the appropriate diet of the species (♀ Epinephelus fuscoguttatus × â™‚ E. lanceolatus) only in scopus I found 18 papers, so the research can be considered important for the scientific community and producers.This should be emphasized in the introductory section.

The methodological structure seems appropriate, the results and discussion section, although separate, is understandable for the readers. 

There are several study variables that cover aspects such as:

1.Antioxidant parameters in serum, 

2.-Histological structure of live, 

3.-Expression of antioxidant and immune-related genes in liver. 

Hybrid grouper, has its market and popular in some regions of the world, the traditional diets as mentioned in the document with high levels of protein, it is correct is changing, the trend in high energy diets, is beginning to use some species, for the economic and environmental benefits that is important to investigate, this aspect should also highlight it with more empirical evidence. For your support I attach the following DOI to be consulted that may be useful.

10.1016/j.fsi.2022.08.013

10.3389/fmars.2022.924018

10.1016/j.fsi.2017.11.007

10.1016/j.fsi.2022.05.016

10.3390/metabo13020305

10.1016/j.aquaculture.2019.01.044

10.1016/j.fsi.2017.10.022

10.1016/j.aqrep.2022.101217

10.1016/j.fsi.2019.02.016

10.1016/j.aquaculture.2022.738088

10.3389/fnut.2022.813249

10.1016/j.fsi.2019.02.052

10.3389/fmars.2022.990193

10.1039/d1fo04085e

10.1016/j.fsi.2023.108815

10.1016/j.fsi.2022.03.038

10.1016/j.aquaculture.2021.737453

10.1016/j.fsi.2019.09.034

Specific comments: 

The title with respect to the objective is suggested should be aligned as much as possible, when writing the objective in the text is short the expression that the authors used , from the methodological point of view, every research idea is born from the research question, to later structure the objective and Hypothesis, I consider that this objective should consider the study variables and the effect of the high energy diet with different levels of protein. 

A more appropriate objective to those performed in the research is: 

The objective of the present research was to determine the effect of high energy diets with different protein levels on Growth performance, Antioxidant parameters in serum, Histological structure of live, expression of antioxidant and immune-related of the hybrid grouper (Epinephelus fuscoguttatus × Epinephelus lanceolatus). 

This objective that I wrote in the previous paragraph, was in accordance with the study variables that are in the results section of their article, which can be shortened in agreement with the authors.

The objective must coincide in the abstract and in the text of the introductory section and must be related to the title.

1.-The title:

The Title, is the section that attracts the attention of the readers.

From the epistemological point of view a title is recommended: 

The object of study: the variables 

The subject of study: who lives the problem.

The method: which in this case is (the effect of) but can be Relate, Evaluate, Impact of... correlation, relationship etc. It depends on the type of research.

With these elements I hope it can help you to make the title a little shorter and more attractive. 

I hope you take it as a constructive idea, the freedom of naming the work is only of the authors.

Effect of high lipid dietary protein ratio on growth, antioxidant parameters, histological structure and expression of antioxidant and immune-related of Hybrid Grouper (22 words)

Or as indicated by its objective in the abstract, just transform it to title:  

Effects of dietary protein level in high lipid diets on serum, liver biochemistry, liver histology e limmune, antioxidant indexes and gene mRNA expression of juvenile hybrid grouper (28 words).

This title is in accordance with the general objective, is short and expresses the study variables.

The object of the study: Growth, Antioxidant Parameters, Histological Structure and Expression of Antioxidant and Immune-Related

Subject of study: Hybrid Grouper  

Method: Effect of

2.-The abstract 

The abstract is well, it is recommended, to be structured in sections, with a brief introduction about the field of study, objective of the research, methodology, results and conclusion. The reader should be able to observe where a section begins and ends. 

For example: 

The Hybrid Grouper is ............. The diets ............. The problem ...........................

Therefore, the objective of the research is ..............

Methodologically, ................................................. was carried out.

The results indicated .........................

It is concluded that......................................................

3-. Simple  Summary

a clear statement of the problem addressed, the aims and objectives, pertinent results, conclusions from the study and how they will be valuable to society.

Write in order as indicated in the author's guide.

4.-Keywords:

Epinephelusfuscoguttatus × â™‚ E. lanceolatus; high-lipid diet; dietary protein level; an-tioxidant; non-specific immunity

The authors' guidelines indicate that keywords should be specific to the article, but reasonably common within the subject discipline. in addition, I suggest not repeating those already included in the title, nor the abstract. 

For example: 

Hybrid Grouper Aquaculture, Animal Feed, aquaculture Diets.

5.-Introduction 

In the introduction there are three paragraphs, the indications in the author's guide must be followed, it is important that this section offers the reader the necessary information where the field of knowledge is highlighted, the importance of the hybrid, the problem statement with its justification, and it is advisable to mention the knowledge gap, which will support the originality of your research, because it is pertinent and relevant.

In an introduction it is appropriate that it should contain:

1. Is it focused on the problem? 

2. Is it pleasant and fruitful to read? Does it summarize the letters and is it prolific in ideas? 

3. Does it clearly state the reasons for conducting the study? 

4. Does it state the premises on which the study is based?

 5. Does it clearly define the objectives of the study? 

6. Does it state the hypotheses that the study intends to demonstrate?

The authors' guide states; 

The introduction should briefly place the study in a broad context and highlight why it is important. It should define the purpose of the work and its importance, including the specific hypotheses being tested. It should carefully review the current state of the field of research and cite key publications. Highlight controversial and divergent hypotheses when necessary. Finally, briefly mention the main objective of the paper. Keep the introduction understandable to scientists working outside the topic of the article.

So, it is not bad, what you write, but it should be improved to support the reader to continue reading, what is most read in an article is the abstract and the introduction. Therefore, I encourage you to improve it.

The keyword hybrid grouper in Google Trends shows the search trend in Google for the term and the countries where most people search for information. I invite you to explore it.

6.-Methodological inaccuracies: 

Recommendations for methodology.

These questions are suggested to be addressed personally, they are recommendations that are useful for the authors.

Did you provide all the necessary information about the populations studied and the products used (doses, origin)? 

 Did you include all the methods used in the study?

 Did you describe them in detail? 

Did you correctly cite the methods? 

Are the statistical procedures rigorous?

Is the use of parametric and non-parametric methods consistent in the description of data and their statistical treatment?

These are reflection questions that support your work.

1.-The first section of the methodology I recommend is usually the study site.

2.- Line 89 

What is the difference between the formulated diets and the commercial diets? 

3.-The diets you formulated are based on fishmeal, it would be very interesting to indicate in an additional column in Table 1. the ingredient in % so that the reader can easily see the differences.

2.-the water conditions, do you mean at the time of sowing or are they an average of the study period, please clarify.

3.-I will preferably indicate the stocking density for the beginning of the experiment.

5.-I recommend being more specific with respect to the variables that were compared with the test. Line 168

6.-In all tables please clarify which is the control diet.

The authors are strongly recommended that their methodology be clear and replicable, hence the importance of reviewing the empirical evidence and similar works.

7.-Results 

Recommendation for results for your support.

Is the parallelism between the presentation of results in the text and the presentation of data in tables and figures perfect? 

 Does the order of presentation of the different types of results follow a logical order? 

Have you highlighted the star results?

 Is it clear in all comparisons which values are compared and which test is used for comparison? 

Is the use of descriptive parameters and tests consistent with the sample sizes and type of data distribution? 

Do you provide the p value in the text or illustrations when the test is significant? 

Can you present the data in a more concise way?

Comments are added to the PDF text of the manuscript.

8.-Discussion 

Recommendations for discussion of the results:

Do you begin by presenting the answer to the main question stated in the introduction? 

Does it elaborate on applications or implications of your answer? 

 Does it highlight the novelty of the work by explaining what the conclusions reached add to existing knowledge? 

Do you claim priority if appropriate? 

Do you explain why the answer follows from the results, why it is reasonable, and how it fits within existing knowledge? 

Do you use scientific hypotheses rigorously? 

Do you not reiterate the results? 

Is the manuscript clear, relevant to the field, and presented in a well-structured manner?

These are questions for your reflection and support.

Is there sufficient empirical evidence for your discussion?

Comments are added to the PDF text of the manuscript.

1.-comparing and discussing results with other species is not appropriate, you should strive to find empirical evidence with respect to the species under study.

Mero hibrido and Diets

9.-Conclusions, Recommendation for conclusions:

1.-Degree of linkage with the objectives. 

2.-Degree of integration of the theoretical and application framework. 

3.-Discussion raised with respect to the results obtained. 

4.-Derivation of normative or explanatory processes on reality. 

5.-Clarification of the limits of the study and proposals for further studies.

These are questions for reflection and support.

The conclusion begins with the phrase, in summary, is not correct,

Its conclusion must show the contribution to knowledge, in accordance with the research objectives, it must give an answer to the research objectives.

They start with the regression model, which is not an objective, it is part of the work, but it was not the central axis, it is a derivation of it.

A conclusion should not hypothesize, on the contrary, it should accept or reject hypotheses.

The correct is: the findings suggest that .......

or with the theoretical and empirical evidence of the present work it is possible to affirm that.....................

and should go in order according to the study variables.

The conclusion I suggest that it should be restated, note that they have many study variables should suggest future lines of research.

I invite the authors to respond to the comments point by point. Preferably with a letter indicating the comment, the response and the change in the manuscript.

When authors disagree, they should provide a clear response with theoretical or empirical evidence.

Author Response

Respond to review: Thank you very much for taking the time to review my manuscript and give me your valuable comments. I responded according to the reviewer's comments in each chapter

REVIEW

The title with respect to the objective is suggested should be aligned as much as possible, when writing the objective in the text is short the expression that the authors used, from the methodological point of view, every research idea is born from the research question, to later structure the objective and Hypothesis, I consider that this objective should consider the study variables and the effect of the high energy diet with different levels of protein.

A more appropriate objective to those performed in the research is: 

The objective of the present research was to determine the effect of high energy diets with different protein levels on Growth performance, Antioxidant parameters in serum, Histological structure of live, expression of antioxidant and immune-related of the hybrid grouper (Epinephelus fuscoguttatus × Epinephelus lanceolatus). 

This objective that I wrote in the previous paragraph, was in accordance with the study variables that are in the results section of their article, which can be shortened in agreement with the authors.

The objective must coincide in the abstract and in the text of the introductory section and must be related to the title.

1.-The title:

The Title, is the section that attracts the attention of the readers.

From the epistemological point of view a title is recommended: 

The object of study: the variables 

The subject of study: who lives the problem.

The method: which in this case is (the effect of) but can be Relate, Evaluate, Impact of... correlation, relationship etc. It depends on the type of research.

With these elements I hope it can help you to make the title a little shorter and more attractive.

I hope you take it as a constructive idea, the freedom of naming the work is only of the authors.

Effect of high lipid dietary protein ratio on growth, antioxidant parameters, histological structure and expression of antioxidant and immune-related of Hybrid Grouper (22 words)

Or as indicated by its objective in the abstract, just transform it to title:  

Effects of dietary protein level in high lipid diets on serum, liver biochemistry, liver histology e limmune, antioxidant indexes and gene mRNA expression of juvenile hybrid grouper (28 words).

This title is in accordance with the general objective, is short and expresses the study variables.

The object of the study: Growth, Antioxidant Parameters, Histological Structure and Expression of Antioxidant and Immune-Related

Subject of study: Hybrid Grouper  

Method: Effect of

Respond:Thank for you comments. I have already corrected the title.

2.-The abstract 

The abstract is well, it is recommended, to be structured in sections, with a brief introduction about the field of study, objective of the research, methodology, results and conclusion. The reader should be able to observe where a section begins and ends. 

For example: 

The Hybrid Grouper is ............. The diets ............. The problem ...........................

Therefore, the objective of the research is ..............

Methodologically, ................................................. was carried out.

The results indicated .........................

It is concluded that......................................................

Respond: Thank you for your suggestions. I have revised the abstract according to your suggestion

3-. Simple Summary

a clear statement of the problem addressed, the aims and objectives, pertinent results, conclusions from the study and how they will be valuable to society.

Write in order as indicated in the author's guide.

Respond: Thanks for your suggestion. I have modified the simple summary according to your suggestion, and summarized the purpose, results and social value of the experiment

4.-Keywords:

Epinephelusfuscoguttatus × â™‚ E. lanceolatus; high-lipid diet; dietary protein level; an-tioxidant; non-specific immunity

The authors' guidelines indicate that keywords should be specific to the article, but reasonably common within the subject discipline. in addition, I suggest not repeating those already included in the title, nor the abstract. 

For example: 

Hybrid Grouper Aquaculture, Animal Feed, aquaculture Diets.

Respond: Thanks for your suggestion, I have modified the keywords

5.-Introduction 

In the introduction there are three paragraphs, the indications in the author's guide must be followed, it is important that this section offers the reader the necessary information where the field of knowledge is highlighted, the importance of the hybrid, the problem statement with its justification, and it is advisable to mention the knowledge gap, which will support the originality of your research, because it is pertinent and relevant.

In an introduction it is appropriate that it should contain:

  1. Is it focused on the problem? 
  2. Is it pleasant and fruitful to read? Does it summarize the letters and is it prolific in ideas? 
  3. Does it clearly state the reasons for conducting the study? 
  4. Does it state the premises on which the study is based?
  5. Does it clearly define the objectives of the study? 
  6. Does it state the hypotheses that the study intends to demonstrate?

The authors' guide states; 

The introduction should briefly place the study in a broad context and highlight why it is important. It should define the purpose of the work and its importance, including the specific hypotheses being tested. It should carefully review the current state of the field of research and cite key publications. Highlight controversial and divergent hypotheses when necessary. Finally, briefly mention the main objective of the paper. Keep the introduction understandable to scientists working outside the topic of the article.

So, it is not bad, what you write, but it should be improved to support the reader to continue reading, what is most read in an article is the abstract and the introduction. Therefore, I encourage you to improve it.

The keyword hybrid grouper in Google Trends shows the search trend in Google for the term and the countries where most people search for information. I invite you to explore it.

Respond: Thanks for your suggestion, we have revised the introduction

6.-Methodological inaccuracies:

Did you provide all the necessary information about the populations studied and the products used (doses, origin)?

 Did you include all the methods used in the study?

 Did you describe them in detail? 

Did you correctly cite the methods? 

Are the statistical procedures rigorous?

Is the use of parametric and non-parametric methods consistent in the description of data and their statistical treatment?

These are reflection questions that support your work.

1.-The first section of the methodology I recommend is usually the study site.

2.- Line 89 

What is the difference between the formulated diets and the commercial diets? 

3.-The diets you formulated are based on fishmeal, it would be very interesting to indicate in an additional column in Table 1. the ingredient in % so that the reader can easily see the differences.

2.-the water conditions, do you mean at the time of sowing or are they an average of the study period, please clarify.

3.-I will preferably indicate the stocking density for the beginning of the experiment.

5.-I recommend being more specific with respect to the variables that were compared with the test. Line 168

6.-In all tables please clarify which is the control diet.

The authors are strongly recommended that their methodology be clear and replicable, hence the importance of reviewing the empirical evidence and similar works.

Respond: Regarding the products used in this experiment, such as the enzyme activity kit, I have already explained his source and other necessary information. We rechecked the manuscript and revised it according to your comments. I have made additional explanations about the methodological steps, and if more detailed instructions are needed, I can upload them in the appendix.

  1. Thanks for the suggestion, I've made the changes based on your comments.
  2. The formulated diet is a high-fat diet with a fat content of 16% and a protein content of about 50% (Suo et al., 2022), which we have simulated based on previous experiments. The commercial diet is a diet with a protein level of 50% and a fat level of 10%. The difference is the fat level.’
  3. Thank you for your suggestion, I have changed the units to percentages in Table 1
  4. The state of the water condition is after the experiment has been carried out and during the research period
  5. The stocking density at the beginning of the experiment I have added in the manuscript.
  6. Thanks for the advice. We have made changes and added to the manuscript
  7. Thank you for your advice. I have stated in the abstract section as well as in the section on feed preparation that the control diet is 510 g/kg group, if I change this in all the tables, some of the table content will be in a smaller font, which may interfere with the reader's view.

7.-Results 

Recommendation for results for your support.

Is the parallelism between the presentation of results in the text and the presentation of data in tables and figures perfect?

Respond: Thanks for the advice. We think it's perfect.

 Does the order of presentation of the different types of results follow a logical order? 

Respond: We have changed the order of presentation of the results according to your suggestions, so that it is more logical order

Have you highlighted the star results?

Respond: The significance of the data I have superscripted with different letters in the table.

Is it clear in all comparisons which values are compared and which test is used for comparison?

Respond: Thank you for your advice. We have added in the manuscript

Is the use of descriptive parameters and tests consistent with the sample sizes and type of data distribution?

Respond: Thank you for your question. The use of descriptive parameters and tests is consistent with sample size and type of data distribution

Do you provide the p value in the text or illustrations when the test is significant?

Respond: We have added P-values to the tables.

Can you present the data in a more concise way?

Respond: Gene expression data we use figures to show trends that are very intuitive. Enzyme activity data we use tables to show because the expression or unit of different enzyme activities are not consistent, and it is difficult to standardise the unit by using a graph. This is a more concise way of presenting the data that we have considered.

Comments are added to the PDF text of the manuscript.

8.-Discussion

Recommendations for discussion of the results:

Do you begin by presenting the answer to the main question stated in the introduction?

Does it elaborate on applications or implications of your answer?

Does it highlight the novelty of the work by explaining what the conclusions reached add to existing knowledge?

Do you claim priority if appropriate?

Do you explain why the answer follows from the results, why it is reasonable, and how it fits within existing knowledge?

Do you use scientific hypotheses rigorously?

Do you not reiterate the results?

Is the manuscript clear, relevant to the field, and presented in a well-structured manner?

These are questions for your reflection and support.

Is there sufficient empirical evidence for your discussion?

Comments are added to the PDF text of the manuscript.

1.-comparing and discussing results with other species is not appropriate, you should strive to find empirical evidence with respect to the species under study.

Respond: We started from the main question posed in the introduction and explained in detail the application of high-fat feeds. This experiment was supplemented with the most suitable protein levels in the context of high-fat feeds. It is justified because high-fat feeds save a certain amount of protein as an energy feed. There is sufficient empirical evidence for our discussion.

Respond: If the research basis of the protein level of hybrid grouper (Epinephelus fuscoguttatus × Epinephelus lanceolatus).in high-fat diet is too much, and the discussion experience is too much, our experiment will become not much reference significance. We introduced grouper, catfish, shrimp, etc. into the discussion, which is more convincing.

9.-Conclusions,

Recommendation for conclusions:

1.-Degree of linkage with the objectives. 

2.-Degree of integration of the theoretical and application framework. 

3.-Discussion raised with respect to the results obtained. 

4.-Derivation of normative or explanatory processes on reality. 

5.-Clarification of the limits of the study and proposals for further studies.

These are questions for reflection and support.

The conclusion begins with the phrase, in summary, is not correct,

Its conclusion must show the contribution to knowledge, in accordance with the research objectives, it must give an answer to the research objectives.

They start with the regression model, which is not an objective, it is part of the work, but it was not the central axis, it is a derivation of it.

A conclusion should not hypothesize, on the contrary, it should accept or reject hypotheses.

The correct is: the findings suggest that .......

or with the theoretical and empirical evidence of the present work it is possible to affirm that.....................

and should go in order according to the study variables.

The conclusion I suggest that it should be restated, note that they have many study variables should suggest future lines of research.

I invite the authors to respond to the comments point by point. Preferably with a letter indicating the comment, the response and the change in the manuscript.

When authors disagree, they should provide a clear response with theoretical or empirical evidence.

Respond: Thank you for your suggestions. Regarding the conclusion section, we have revised it based on your comments, so please consult the manuscript!

Reference:

Suo, X., Yan, X., Tan, B., Pan, S., Li, T., Liu, H., et al. (2022). Lipid metabolism disorders of hybrid grouper (♀Epinephelus fuscointestinestatus × â™‚E. lanceolatu) induced by high-lipid diet. Front. Mar. Sci. 9, 1–15. doi: 10.3389/fmars.2022.990193.

Reviewer 2 Report

Comments and Suggestions for Authors

This manuscript tested the effects of dietary protein level in high -lipid diets on serum and liver biochemistry, liver histology, and liver immune and antioxidant indexes and gene mRNA expression of juvenile hybrid grouper (Epinephelus fuscoguttatus ×E. lanceolatus). For fish nutrition and feed research, it is a routine study. For hybrid grouper, it enriches the nutritional parameters and mechanism explanation. Data obtained may support grouper culture. However, the manuscript needs minor revision and several considerations are described below.

Q1: Line 101, please provide the protein and fat levels of commercial feed.

Q2: How did the authors determine the satiation status of the juvenile fish? Will there be sinking feed at feeding time? How to deal with it?

Q3: Line 124-125, “After sampling, all samples are stored at -80 °C”, do sectioned samples need to be stored at -80°C? Is there a wrong way to do this?

Q4: Line 129, formatting error with two spaces between fonts. There are a number of other formatting errors throughout the text, not all of which are listed here, so the author is asked to check and correct them carefully.

Q5: Line 134-137, please provide the national standard method for testing feed moisture, protein and fat.

Q6: Line 154, sections 5 μm thickness, please provide references for support

Q7: Table 2, please provide full names of primer genes

Q8: Table 5, formatting is not consistent with other tables and discrepancies are not superscripted, please correct.

Q9: Is beta actin the best house keeping genes? although its widely used, but its efficiency in some tissues/species have been questione.

Q10:There are multiple spelling, grammatical errors throughout the manuscript.

Comments on the Quality of English Language

There are multiple spelling, grammatical errors throughout the manuscript.

Author Response

Respond: Thank you for taking the time to review our manuscript. We have revised it according to your comments, and the revised manuscript has been uploaded.

This manuscript tested the effects of dietary protein level in high -lipid diets on serum and liver biochemistry, liver histology, and liver immune and antioxidant indexes and gene mRNA expression of juvenile hybrid grouper (Epinephelus fuscoguttatus ×E. lanceolatus). For fish nutrition and feed research, it is a routine study. For hybrid grouper, it enriches the nutritional parameters and mechanism explanation. Data obtained may support grouper culture. However, the manuscript needs minor revision and several considerations are described below.

Respond: Thank you very much for your comments and revisions to my manuscript. I have revised or supplemented it in the manuscript according to your opinions, and I have also responded to some questions

Q1: Line 101, please provide the protein and fat levels of commercial feed.

Respond: Thanks for your advice, I have added the details of commercial feed to the manuscript

Q2: How did the authors determine the satiation status of the juvenile fish? Will there be sinking feed at feeding time? How to deal with it?

Respond: When feeding is carried out, the grouper no longer compete for food and gradually swim to the bottom of the tank, which we think is a full state. When there is a feed bottom, we will fish out and air dry, and weigh records for later FCR calculations

Q3: Line 124-125, “After sampling, all samples are stored at -80 °C”, do sectioned samples need to be stored at -80°C? Is there a wrong way to do this?

Respond: Thanks for your comments, I have corrected my wrong description

Q4: Line 129, formatting error with two spaces between fonts. There are a number of other formatting errors throughout the text, not all of which are listed here, so the author is asked to check and correct them carefully.

Respond: Thank you, I have carefully checked the manuscript and revised it

Q5: Line 134-137, please provide the national standard method for testing feed moisture, protein and fat.

Respond: Thank you. Details have been added in the manuscript

Q6: Line 154, sections 5 μm thickness, please provide references for support

Respond: Thank you for your comments. We have added the corresponding references

Q7: Table 2, please provide full names of primer genes

Respond: Thank you for your advice. I've added the full name of the primer gene to the manuscript

Q8: Table 5, formatting is not consistent with other tables and discrepancies are not superscripted, please correct.

Respond: Thanks for your comments. I have corrected.

Q9: Is beta actin the best housekeeping genes? although its widely used, but its efficiency in some tissues/species have been question.

Response: We would like to thank you for your careful reading, helpful comments, and constructive suggestions, which have significantly improved the presentation of our manuscript. As you said, the beta actin housekeeping gene is indeed controversial and is expressed with varying efficiency in different tissues/species. But the reason we chose this gene is also a combination of considerations. In Liang's study, the housekeeping genes of grouper were screened and the stability of 17 housekeeping genes in 11 tissues of hybrid grouper was assessed by using different mathematical algorithms (Genorm, normFinder, BestKeeper and comparative ΔCt method). In the conclusion of this study, it was also concluded that in liver tissues, action the housekeeping gene was more stable (Liang et al., 2022).

Q10: There are multiple spelling, grammatical errors throughout the manuscript.

There are multiple spelling, grammatical errors throughout the manuscript.

Response: Thank you for your advice. I have checked the full text and fixed some syntax errors

Refrence

Liang, J., Xu, J., Xie, S., Cao, J., & Tan, B. (2022). Selection of Appropriate Housekeeping Genes for Gene Expression Normalization in Hybrid Grouper (Epinephelus fuscoguttatusâ™€× E. lanceolatus♂). Turkish Journal of Fisheries and Aquatic Sciences, 22(9). doi:10.4194/TRJFAS20646

Round 2

Reviewer 2 Report

Comments and Suggestions for Authors

The authors have answered all my questions. I have no more questions.

Comments on the Quality of English Language

The language has met the requirements for publication.